# LEARNING SUCCESSOR REPRESENTATIONS WITH DISTRIBUTED HEBBIAN TEMPORAL MEMORY

## ABSTRACT

This paper presents a novel approach to address the challenge of online hidden representation learning for decision-making under uncertainty in non-stationary, partially observable environments. The proposed algorithm, Distributed Hebbian Temporal Memory (DHTM), is based on factor graph formalism and a multicomponent neuron model. DHTM aims to capture sequential data relationships and make cumulative predictions about future observations, forming Successor Representation (SR). Inspired by neurophysiological models of the neocortex, the algorithm utilizes distributed representations, sparse transition matrices, and local Hebbian-like learning rules to overcome the instability and slow learning process of traditional temporal memory algorithms like RNN and HMM. Experimental results demonstrate that DHTM outperforms classical LSTM and performs comparably to more advanced RNN-like algorithms, speeding up Temporal Difference learning for SR in changing environments. Additionally, we compare the SRs produced by DHTM to another biologically inspired HMM-like algorithm, CSCG. Our findings suggest that DHTM is a promising approach for addressing the challenges of online hidden representation learning in dynamic environments.

## 1 INTRODUCTION

Modelling sequential data is one of the most important tasks in Artificial Intelligence as it has many applications, including decision-making and world models, natural language processing, conversational AI, time-series analysis, and video and music generation (Min et al., 2021; Eraslan et al., 2019; Dwivedi et al., 2023; Ji et al., 2020; Moerland et al., 2023). One of the classical approaches to modelling sequential data is forming a representation that stores and condenses the most relevant information about a sequence, and finding a general transformation rule of this information through the dimension of time (Lipton et al., 2015; Harshvardhan et al., 2020; Mathys et al., 2011). We refer to the class of algorithms that use this approach as Temporal Memory (TM) algorithms, as they essentially model the cognitive ability of complex living organisms to remember the experience and make future predictions based on this memory (Hochreiter & Schmidhuber, 1997; Friston et al., 2016; 2018; Parr & Friston, 2017).

This paper addresses the problem of hidden representation learning for decision-making under uncertainty, which can be formalized as agent Reinforcement Learning (RL) for a Partially Observable Markov Decision Process (POMDP) (Poupart, 2005). Inferring the hidden state in a partially observable environment is, in effect, a sequence modelling problem as it requires processing a sequence of observations to get enough information about hidden states. One of the most efficient representations of the hidden states for discrete POMDP is the Successor Representation (SR) that disentangles hidden states and goals given by the reward function (Dayan, 1993). An extension of the SR into continuous POMDP is the Successor Features framework, which employs the same idea of value function decomposition, but, instead, for features of a hidden state (Barreto et al., 2017). Temporal Memory algorithms can be leveraged to make cumulative predictions about future states and their features to form SR or SF.

The most prominent TM algorithms, like a Recurrent Neural Network (RNN) or a Hidden Markov Model (HMM), use backpropagation to capture data relationships, which is known for its instability due to recurrent non-linear derivatives. They also require having complete sequences of data at hand during the training. Although the gradient vanishing problem can be partially circumvented in a

way Receptance Weighted Key Value (RWKV) (Peng et al., 2023) or Linear Recurrent Unit (LRU) (Orvieto et al., 2023) models do, the problem of online learning is still a viable topic. In contrast to HMM, RNN models and their descendants also lack a probabilistic theory foundation, which is beneficial for modeling sequences captured from stochastic environments (Salaün et al., 2019; Zhao et al., 2020). There is little research on TM models that can be used in fully online adaptable systems interacting with partially observable stochastic environments with access only to one sequence data point at a time, a prevalent case in Reinforcement Learning (Jahromi et al., 2022).

We propose a Distributed Hebbian Temporal Memory (DHTM) algorithm based on the factor graph formalism and multi-compartment neuron model. The resulting graphical structure of our model is similar to one of the Factorial-HMM (Ghahramani & Jordan, 1995), but with a factor graph forming online during training. We also show that depending on the graphical structure, our TM can be viewed as an HMM version of either RNN or LRU regarding information propagation in time. An important feature of our model is that transition matrices for each factor are stored as different components (segments) of artificial neurons, which makes computations very efficient in the case of sparse transition matrices. Our TM forms sequence representations fully online and employs only local Hebbian-like learning rules (Hebb, 2005; Churchland & Sejnowski, 1992; Lillicrap et al., 2020), circumventing gradient drawbacks and making the learning process much faster than gradient methods.

Some key ideas for our TM algorithm are inspired by neurophysiological models of the neocortex neural circuits and pyramidal neurons (George & Hawkins, 2009; Hawkins & Ahmad, 2016; O'Reilly et al., 2021). For example, emission matrices for random variables are fixed to resemble the columnar structure of the neocortex layers, which significantly lessens the number of trainable parameters, speeding up learning and leading to sparse transition matrices. Another example is using multi-compartmental neurons with active dendritic segments as independent detectors of neuron pattern activity (London & Häusser, 2005). Each dendritic segment can be viewed as a row of an HMM state transition matrix or, more generally, a value of a discrete factor function. Thus, we don't explicitly store large transition matrices, only their non-zero parts.

The DHTM model notoriously fits Successor Features in the Reinforcement Learning setup to speed up TD learning. The proposed TM is tested as a world model (Ha & Schmidhuber, 2018; Hafner et al., 2023) for an RL agent architecture, making decisions in a simple Pinball-like environment and in a more challenging AnimalAI testbed (Crosby et al., 2020). Our algorithm outperforms a classic RNN algorithm LSTM and a more advanced RNN-like transformer algorithm RWKV in online Successor Feature formation task due to combination of fast Hebbian-like learning and sparse hidden state coding. Another advantage of our algorithm is that it allows its implementation for neuromorphic processors, as it uses only local learning rules.

Our contribution in this work is the following:

- We propose a distributed memory model DHTM based on the factor graph formalism and multicompartment neural model.
- Our model stores sparse factor functions in neural segments, which significantly lessens the number of trainable parameters and speeds up learning.
- The DHTM learns fully online employing only local Hebbian-like rules.
- The DHTM model fits Successor Features in the RL setup to speed up TD learning.
- Tested as a world model for an RL agent architecture in a Pinball environment, DHTM outperforms LSTM and RWKV in online Successor Features formation task.

## 2 BACKGROUND

This section provides basic information about some concepts necessary to follow the paper.

### 2.1 REINFORCEMENT LEARNING

In this paper, we consider decision-making in a partially observable environment, which is usually formalized as Partially Observable Decision Process (Poupart, 2005). A POMDP is defined as a tuple $\mathcal{M} = (S, A, P, R, O, D, \gamma)$, where $S$—state space, $A$—action space, $P(s, a, s') =$

$Pr(s` \mid s, a)$—transition function, $R(s)$–reward function, O—observation space, $D(a, s`, o) = Pr(o \mid a, s`)$—sensor model and $\gamma \in [1, 0)$—discount factor, given a transition $s, a \to s`$, where $s \in S$, $a \in A$, $o \in O$. If $S, A, O$ are finite, $P, D$ can be viewed as real valued matrices, otherwise, they are conditional density functions. Here we consider deterministic rewards, which depend only on the current state, i.e. $R(s) : S \to \mathbb{R}$.

The task of RL is to find a policy $\pi(a \mid s) : S \times A \to [0, 1]$, which maximizes expected return $G = \mathbb{E}[\sum_{t=0}^{T} \gamma^l R_t]$, where $T$ is an episode length. For value based methods, it is convenient to define optimal policy via Q-function: $Q^\pi(s_t, a_t) = \mathbb{E}[\sum_{l \geq t} \gamma^l R(s_{l+1}) \mid s_t, a_t, \pi]$. For an optimal value function $Q^*$ an optimal policy can be defined as $\pi(a \mid s) = \underset{a}{\operatorname{argmax}} Q^*(s, a)$.

## 2.2 HIDDEN MARKOV MODEL

Partially observable Markov process can be approximated by a Hidden Markov model (HMM) with hidden state space $H$ and observation space $O$. $O$ is the same as in $\mathcal{M}$, but $H$ generally is not equal $S$. Variables $H_t$ represent an unobservable (hidden) approximated state of the environment which evolves over time, and observable variables $O_t$ represent observations that depend on the same time step state $H_t$, and $h_t, o_t$ are corresponding values of this random variables. For the sake of simplicity, we suppose that actions are fully observable and information about them is included into $H_t$ variables. For the process of length $T$ with state values $h_{1:T} = (h_1, \ldots, h_T)$ and $o_{1:T} = (o_1, \ldots, o_T)$, the Markov property yields the following factorization of the generative model:

$$p(o_{1:T}, h_{1:T}) = p(h_1) \prod_{t=2}^{T} p(h_t|h_{t-1}) \prod_{t=1}^{T} p(o_t|h_t). \tag{1}$$

In case of discrete hidden state, a time-independent stochastic transition matrix can be learned with Baum–Welch algorithm (Baum et al., 1970), a variant of Expectation Maximization algorithm. To compute the statistics for the expectation step, it employs the forward-backward algorithm, which is a special case of sum-product algorithm (Kschischang et al., 2001).

## 2.3 SUCCESSOR REPRESENTATION

Successor Representations are such representations of hidden states from which we can linearly infer the state value given the reward function (Dayan, 1993). Here, we assume observation and state spaces are discrete.

$$V(h_t = i) = \mathbb{E}[\sum_{l=0}^{\infty} \gamma^l R_{t+l+1} \mid h_t = i] = \sum_{l=0}^{\infty} \gamma^l \mathbb{E}[R_{t+l+1} \mid h_t = i] =$$

$$= \sum_{l=0}^{\infty} \gamma^l \sum_{j} p(h_{t+l+1} = j \mid h_t = i) R_j = \sum_{j} \sum_{l=0}^{\infty} \gamma^l p(h_{t+l+1} = j \mid h_t = i) R_j = \sum_{j} M_{ij} R_j,$$

where $\gamma$ is a discount factor, vector $\text{SR}(h = i) = \{M_{ij}\}_j$ is a Successor Representation of a state $i$, and $M_{ij} = \sum_{l=0}^{\infty} \gamma^l p(h_{t+l+1} = j \mid h_t = i)$. $R_j$ is a reward for observing the state $j$. That is, SR can be computed by a TM that is able to predict future states. TM algorithms effectively predict observations only for a finite time horizon $T$. Therefore, in order to learn SR, a technique similar to TD learning in standard RL may be employed:

$$\delta_{ij} = \sum_{l=0}^{T} \gamma^l p(h_{t+l+1} = j \mid h_t = i)) + \gamma^{T+1} \sum_{k} M_{kj} p(h_{t+T+1} = k \mid h_t = i) - M_{ij}, \tag{2}$$

$$M_{ij} \leftarrow M_{ij} + \alpha \delta_{ij}, \tag{3}$$

where $\alpha \in (0, 1)$ is a learning rate, $\delta_{ij}$—TD error for SR.

In partially observable environments, however, exact state values are not known, therefore we operate with state distributions or so-called belief states (Poupart, 2005), which are inferred from observations. In that case, state value and SR are functions of hidden state variable distribution (see details in Appendix B).

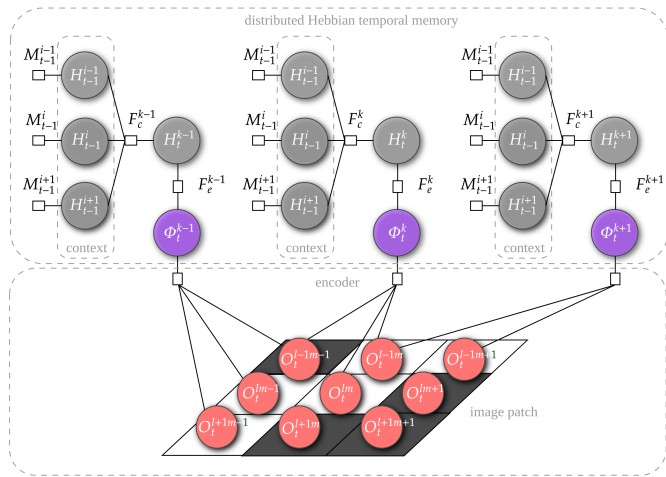

Figure 1: Partial factor graph for the DHTM. The input to the model is a sequence of binary images, each pixel is modelled as Bernoulli random variable $O_t^{lm}$, where $l$ and $m$ denote corresponding rows and cols of the image. The encoder block forms image categorical features $\Phi_t^k$ in an unsupervised manner. Each feature $\Phi$ has its own explaining hidden variable, which may depend on hidden variables of the other features and on itself from the previous time step. $F_c^k$ and $F_e^k$ are context and emission factors for the corresponding variables. Unary factors $M_{t-1}^i$ called messages represent accumulated information about previous time steps.

## 2.4 Sparse Distributed Representations

In our work, we design our model to operate with sparse distributed representations (SDRs) to reflect the spatiotemporal property of cortical network activity (Perin et al., 2011). In the discrete time case, SDR is a sparse binary vector in a high-dimensional space. To encode observed dense binary patterns to SDRs, we use a biologically plausible k-WTA (k-winners take all) neural network algorithm called spatial pooler with a Hebbian-like unsupervised learning method (see details in Appendix A).

## 3 Distributed Hebbian Temporal Memory

### 3.1 Factor Graph Model

Distributed Hebbian Temporal Memory is based on the sum-product belief propagation algorithm in a factor graph (see Figure 1). Analogously to Factorial-HMM (Ghahramani & Jordan, 1997), we divide the hidden space $H$ into subspaces $H^k$. There are four sets of random variables (RV) in the model: $H_{t-1}^i$—latent variables representing hidden states from the previous time step (context), $H_t^k$—latent variables for the current time step, $\Phi_t^k$—feature variables, and $O_t^{lm}$—observable variables. Except for $O_t^{lm}$, all random variables have a categorical distribution. In contrast, $O_t^{lm}$, are Bernoulli variables because they represent pixels from a binary input image observation. RV state values are denoted as corresponding lowercase letters: $h_{t-1}^i$, $h_t^k$, $\varphi_t^k$, $o_t^{lm}$.

Each variable $\Phi_t^k$ is considered independent and has a separate graphical model for increased computational efficiency. However, hidden variables of the same time step are statistically interdependent in practice. We introduce their interdependence through a segment computation trick that goes beyond the standard sum-product algorithm (see Eq. 7).

The model also has three types of factors: $M_{t-1}^i$—messages from previous time steps, $F_c^k$—context factor (generalized transition matrix), $F_e^k$—emission factor. We assume that messages $M_{t-1}^i$ include posterior information from the time step $t-1$, therefore we don't depict observable variables for previous time steps in Figure 1.

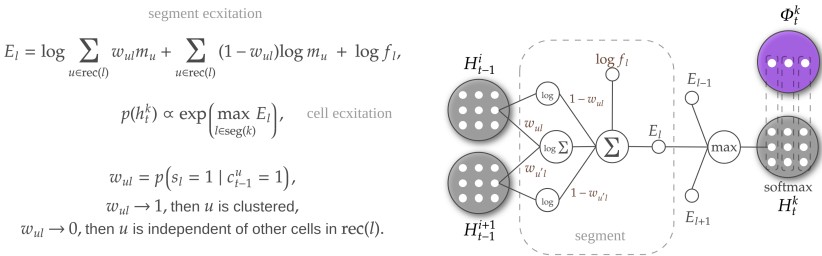

$$E_l = \log \sum_{u \in \text{rec}(l)} w_{ul} m_u + \sum_{u \in \text{rec}(l)} (1 - w_{ul}) \log m_u + \log f_l,$$

$$p(h_t^k) \propto \exp\left(\max_{l \in \text{seg}(k)} E_l\right),$$

$$w_{ul} = p\left(s_l = 1 \mid c_{t-1}^u = 1\right),$$
$w_{ul} \to 1$, then $u$ is clustered,
$w_{ul} \to 0$, then $u$ is independent of other cells in $\text{rec}(l)$.

Figure 2: Neuronal implementation of the DHTM. Random variables are represented by cell clusters (white circles), where each cell corresponds to a state and its spike frequency—to the probability of the state $p(h_t^k)$. Cell's dendritic segments $seg(k)$ correspond to context factor values $f_l$ for a particular combination of states (active presynaptic cells) $rec(l)$. Segments' excitations $E_l$ are combined to determine cell's spike frequency $p(h_t^k)$. Segment's synaptic weights reflect specificity of $rec(l)$ combination for the segment. Emission factors $F_e^k$ are fixed and represented by minicolumns inside a variable.

Further, we discuss only the upper block of the graph, which is DHTM itself. The lower block—an encoder—is described in the Appendix A. The only requirement for the encoder is that its output should be represented as states of categorical variables (features) for the current observation.

## 3.2 NEURAL IMPLEMENTATION

The main routine of the DHTM is to estimate distributions of currently hidden state variables given by the equation 4, the computational flow of which is schematically depicted in Figure 2:

$$p(h_t^k) \propto \sum_{\{h_{t-1}^i : i \in \omega_k\}} \prod_{i \in \omega_k} M_{t-1}^i(h_{t-1}^i) F_c^k(h_t^k, \{h_{t-1}^i : i \in \omega_k\}), \tag{4}$$

where $\omega_k = i_1, \ldots, i_n$—set of previous time step RV indexes included in $F_c^k$ factor, $(n+1)$—factor size.

For computational purposes, we translate the problem to the neural network architecture with Hebbian-like learning (for biological interpretation of the model, see Appendix C). As can be seen from Figure 2, every RV can be viewed as a set of spiking neurons representing the RV's states, that is, $p(h_t^k) = p(c_t^j = 1)$, where $j$—index of a neuron corresponding to the state $h_t^k$. Cell activity is binary $c_t^j \in \{0, 1\}$ (spike/no-spike), and the probability might be interpreted as a spike rate. Factors $F_c^k$ and $M_{t-1}^i$ can be represented as vectors, where elements are factor values for all possible combinations of RV states included in the factor. Let's denote elements of the vectors as $f_l$ and $m_u$ correspondingly, where $l$ corresponds to a particular combination of $k, h_t^k, h_{t-1}^{i_1}, \ldots, h_{t-1}^{i_{n_l}}$ state values and $u$ indexes all neurons representing states of previous time step RVs.

Drawing inspiration from biological neural networks with active dendrites, we group a neuron's connections (dendrites) into segments. A segment acts as an independent computational unit that detects a particular input pattern (a context state) defined by its own receptive field. In our model, a segment links together factor value $f_l$, the computational graph shown in Figure 2, and the excitation $E_l$ induced by the segment $l$ to the cell it is attached to. The segment is active, i.e., $s_l = 1$ if all its presynaptic cells are active; otherwise, $s_l = 0$. Computationally, a segment transmits its factor value $f_l$ to a cell it is attached to if the context matches the corresponding state combination.

We can now rewrite equation 4 as the following:

$$p(h_t^k) \propto \sum_{l \in \text{seg}(j)} L_l f_l^k, \tag{5}$$

where $L_l = \prod_{u \in \text{rec}(l)} m_u$ is segment's likelihood as long as messages are normalized, $\text{seg}(j)$—indexes of segments that are attached to cell $j$, $\text{rec}(l)$—indexes of presynaptic cells that constitute receptive field of a segment with index $l$.

Initially, all factor entries are zero, meaning cells have no segments. As learning proceeds, new non-zero connections grouped into segments are grown. In equation 5 we benefit from having sparse factor value vectors because its complexity depends linearly on the amount of non-zero components. And that's usually the case in our model due to one-step Monte-Carlo learning and specific form of emission factors $F_e^k$:

$$F_e^k(h_t^k, o_t^k) = \mathbb{I}[h_t^k \in \text{col}(\varphi_t^k)], \tag{6}$$

where $\mathbb{I}$—indicator function, $\text{col}(\varphi_t^k)$ is a set of hidden states connected to the feature state $\varphi_t^k$ that forms a column. The form of emission factor is inspired by presumably columnar structure of the neocortex and was shown to induce sparse transition matrix in HMM (George et al., 2021).

Segment likelihood $L_l$, resulting from the sum-product algorithm, is calculated as if presynaptic cells are independent. However, it's not usually the case for sparse factors. To take into account, approximately, their interdependence, we substitute the following equation for segment log-likelihood:

$$\log L_l = \log \sum_{u \in \text{rec}(l)} w_{ul} m_u + \sum_{u \in \text{rec}(l)} (1 - w_{ul}) \log m_u - \log n_l, \tag{7}$$

where $w_{pl}$—synapse efficiency or neuron specificity for segment, such that $w_{ul} = p(s_l = 1 | c_{t-1}^u = 1)$, and $n_l$-number of cells in segment's receptive field.

The idea that underlies the formula is to approximate between two extreme cases:

- $p(s_l = 1 | c_{t-1}^u = 1) \to 1$ for all $u$, which means that all cells in the receptive field are dependent and are part of one cluster, i.e., they fire together. In that case, it should be $p(s_l) = m_u$ for any $u$, but we also reduce prediction variance by averaging between different $u$.

- $p(s_l = 1 | c_{t-1}^u = 1) \to 0$ for all $u$ means that presynaptic cells don't form a cluster. In that case, segment activation probability is just a product of the activation probability of each cell.

The resulting equation for belief propagation in DHTM is the following:

$$p(h_t^k) = p(c_t^j = 1) = \underset{j \in \text{cells}[H_t^k]}{\text{softmax}} \left( \max_{l \in seg(j)} (E_l) \right), \tag{8}$$

where $E_l = \log f_l + \log L_l$, $\text{cells}[H_t^k]$—indexes of cells that represent states for $H_t^k$ variable. Here, we also approximate logarithmic sum with $\max$ operation inspired by the neurophysiological model of segment aggregation by cell (Stuart & Spruston, 2015).

The next step after computing $p(h_t^k)$ distribution parameters is to incorporate information about current observations $p(h_t^k | o_t^k) \propto p(h_t^k)\mathbb{I}[h_t^k \in \text{col}(o_t^k)]$. After that, the learning step is performed. The step for closing the loop of our TM algorithm is to assign the posterior for the current step $p(h_t^k | o_t^k)$ to $M_{t-1}^i$.

DHTM learns $f_l$ and $w_{ul}$ weights by Monte-Carlo Hebbian-like updates. First, $h_{t-1}^i$ and $h_t^k$ are sampled from their posterior distributions: $p(h_{t-1}^i | o_{t-1}^i) \propto M_{t-1}^i$ and $p(h_t^k | o_t^k)$ correspondingly. Then $f_l$ is updated according to the segment's $s_l$ and its cell's $c_t^j$ activity so that $f_l$ is proportional to several coincidences $s_l = c_t^j = 1$ during the recent past, i.e., cell and its segment are active at the same time step. It's similar to Baum-Welch's update rule (Baum et al., 1970) for the transition matrix in HMM, which, in effect, counts transitions from one state to another, but, in our case, the previous state (context) is represented by a group of RVs, not just one hidden RV.

Weights $w_{ul}$ are also updated by the Hebbian rule to reflect the specificity of a presynaptic $u$ for activating a segment $l$. That is, they are targeted to represent probability $p(s_l = 1 | c_{t-1}^u = 1)$ that segment $s_l$ is active, given cell $u$ was active at the previous time-step. We could learn it by counting activation coincidences and mismatches. But in our algorithm it is approximated as exponential moving average of segment's $s_l$ frequency activation, given $c_{t-1}^u = 1$: $\Delta w_{ul} = \alpha \cdot \mathbb{I}[c_{t-1}^u = 1] \cdot (\mathbb{I}[s_l = 1] - w_{ul})$, where $\alpha \in [0, 1)$ — learning rate.

### 3.3 AGENT ARCHITECTURE

We incorporate DHTM as a part of an RL agent. The agent consists of a DHTM memory model, an SF mapping from hidden space, and a feature reward function. The memory model aims to speed

---

**Algorithm 1** General agent training procedure

---

1: **for** episode=1..n **do**
2:     RESET_MEMORY()
3:     action ← null
4:     **while** (**not** terminal) **and** (steps < max_steps) **do**
5:         obs, reward ← STEP()
6:         features ← ENCODE(PREPROCESS(obs))
7:         OBSERVE(features, action)
8:         REINFORCE(reward, features)
9:         action ← SAMPLE_ACTION()
10:        ACT(action)
11:    **end while**
12: **end for**

---

up SF learning by predicting cumulative future distributions of feature variables $\Phi$ according to equation 17. As shown in equation 13, SF representations are learned to estimate state value. The $r(\varphi_t^k)$ reward function is also learned during interaction with the environment and, combined with SF representations, is used to estimate the action value function.

The agent training procedure is outlined in Algorithm 1. For each episode, the memory state is reset to a fixed initial message with RESET_MEMORY() and action variable is initialized with null value. An observation image returned by an environment (obs) is first preprocessed to get events, mimicking a simple event-based camera with a floating threshold determined from the average difference between the current and previous step image intensities. The resulting events are encoded to SDRs with a biologically inspired spatial pooling encoder described in Appendix A. In OBSERVE() routine, the memory learns to predict next feature states as described in Section 3 and SF learning happens according to equation 16. An agent learns associations to feature states and rewards in line 8:

$$r_i^k \leftarrow r_i^k + \alpha \mathbb{I}[\varphi_t^k = i](R_t - r_i^k) \tag{9}$$

where $\alpha$ is a learning rate, $R_t$—a reward for the current time step.

We include actions into the model by forcing some of the hidden variables $H_t^k$ to represent actions. That is, we assume that information about action is included in the hidden state of the model. For example, if we have 4 actions, we set 4 states for one of the hidden variables and set its state from observation of the action. We form on-policy SFs, i.e. relying on policy iteration theorem.

An agent has a softmax policy over predicted values: $\pi(a_t \mid o_{0:t}) = \text{softmax}(V[p(h_{t+1} \mid o_{0:t}, a_t)])$. We use the model to predict the hidden state distribution for every action in the next timestep $t + 1$ and then estimate its value according to equation 13.

## 4 EXPERIMENTS

We test our model in a reinforcement learning task in a pinball-like 2D environment, where successor features are easy to interpret, and in a more challenging AnimalAI 3D environment. This section shows how different memory models affect SF learning and an RL agent's adaptability. In our work, we compare the proposed DHTM model with LSTM (Hochreiter & Schmidhuber, 1997), RWKV (Peng et al., 2023), and CSCG (George et al., 2021) (see Appendix E for the details).

### 4.1 PINBALL

The first, classic maze, test is designed in the Pinball environment (see Appendix F for details) to qualitatively assess SFs formed by different TMs for random policy (see Fig. 3). Ball is controlled by the agent able to apply a momentum in four opposite directions. The ball and terminal state are separated by a wall with a door on the right. Each episode is maximum of 30 steps. Memories are tested in two regimes: 5-step planning (i.e. using equation 17 only) and prediction only (equation 18). As can be seen from the heatmaps, only DHTM yields adequate value functions. However, as can be seen from the learning curves, surprise of DHTM is higher than of the other memories. LSTM's learning curve is much flatter than of the others. Five-step DHTM planning gives more

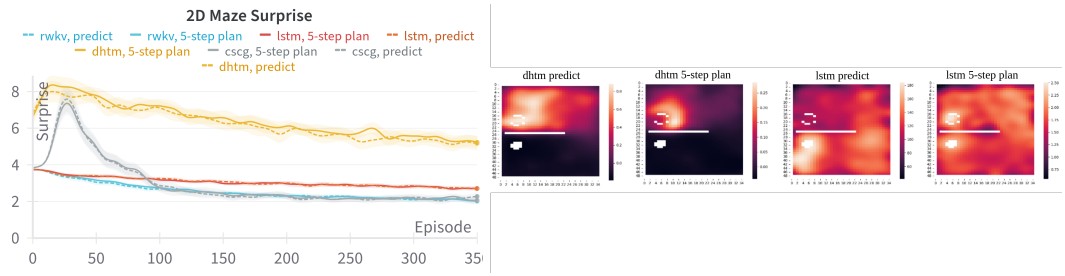

Figure 3: Results of 2D maze random policy experiment in the Pinball environment. Surprise learning curves for DHTM, LSTM, RWKV and CSCG. Heatmaps represent value functions for DHTM and LSTM.

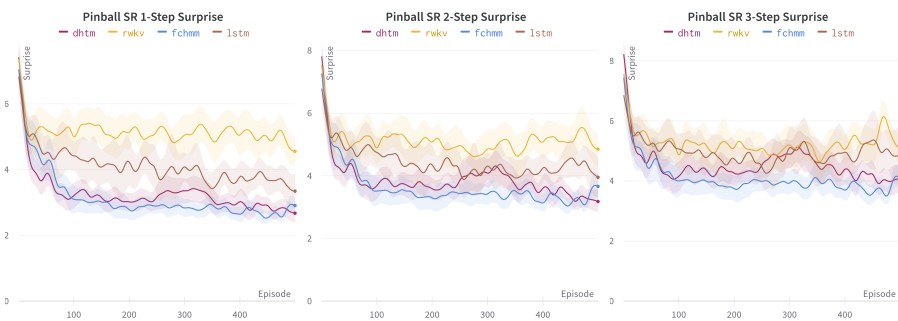

Figure 4: Surprise comparison for various memory models including DHTM (ours), LSTM, RWKV, and Factorial version of CSCG (fchmm). The SFs generated by normalized five-step prediction models are used to calculate surprise for three future time steps.

abrupt value function in comparison to prediction regime, as it usually requires more than five steps to reach the goal. Heatmaps for other baselines can be found in Appendix G.

The second test is to show how TM can enhance adaptation in changing environments. For that experiment, we use two configurations of the Pinball environment shown in Figure 7-A. We narrow the action space to three momentum vectors: vertical, 30 degrees left and 30 degrees right from the vertical axis. Each time step, the agent gets a small negative reward and a large positive reward if the ball enters the force field in the centre. The episode finishes when the ball enters the rewarding force field or the maximum number of steps is reached. Each trial is run for 500 episodes, each a maximum of 15 steps long, and we average the results over three trials for each parameter set and memory model.

We test the accuracy of five-step SF representations by measuring their pseudo-surprise, which is surprise computed for observed states on different time steps after SF was predicted with respect to normalized SF (more details in Appendix D). In all experiments, the encoder outputs five variables Φ with 50 states each. As can be seen from Figure 4, SRs produced by our memory model (dhtm) give lower surprise than SRs of LSTM (lstm) and RWKV (rwkv), and is on par with SRs produced by Factorial version of CSCG (fchmm), which is just several CSCGs trained in parallel to enable handling of multiple variables outputted by encoder.

Then, we test how the number of prediction steps affects the agent's adaptability in the Pinball environment. In the first 500 episodes, the agent is trained to reach the target in the centre, as shown in Figure 7-A, then the target is blocked by a random force that applies force in perpendicular direction to the ball's movement. The results show that an agent that uses five prediction steps during n-step TD learning of SF faster adapts to the changes in the environment in comparison to 1-step TD learning for SF, as seen from Figure 5-A.

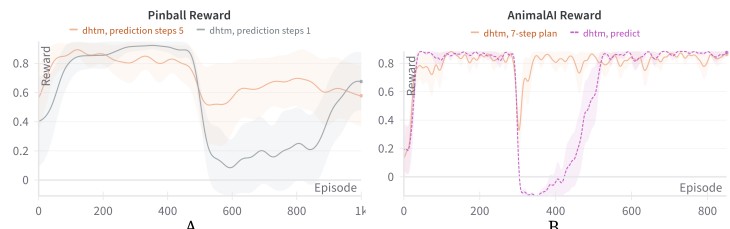

Figure 5: A. Comparison of agent's adaptability during changes in the environment with different prediction steps during n-step TD learning of SF. At the 500th episode, the environment changes its configuration, shown in Figure 7-A. B. AnimalAI changing food position experiment. Left picture is DHTM reward curves each averaged over five trials for two cases: SF formed by 7-step planning using DHTM and SF is predicted using TD learned weights and DHTM inferred belief states. At the 300th episode, the food is moved to the opposite corridor (see Fig. 7-C).

## 4.2 ANIMALAI

We designed an experiment in AnimalAI environment shown on Figure 7-C. There are two corridors, one of which contains food (yellow cirle). The agent makes a decision at the start of the trial, having three options: go to the left corridor, go to the right and stay turning. After the decision is made, the agent follows a fixed strategy, which brings it either to the right corridor or to the left, and it observes its movement and actions. An episode ends when strategy is executed. Each time step, agent gets small negative reward and big positive reward only if reaches food. After 300 episodes, food is placed to the other corridor. Reward curves averaged over five trials for each setup are presented in Figure 5-B. There are two cases on the plot: SF is formed by prediction (equation 18) or planned (equation 17). The results for DHTM show that planning allows much faster adaptation to the change of the rewarding food position.

## 5 CONCLUSION

In this paper, we introduce a novel probabilistic Factorial-HMM-like algorithm DHTM for learning an observation sequence model in stochastic environments that uses local Hebbian-like learning rules, which renders it apt for running on neuromorphic processors. DHTM is scalable to multiple feature variables as it employs sparse distributed representations and sparse factor function implementation using segments, which biologically plausible multicomponent neural models inspire. In contrast to methods that use Monte-Carlo trajectory sampling for future states probability estimation, our method is able to perform belief propagation, so each prediction step adds constant amount of computations. We show that our memory model can quickly learn the observation sequences representation and the transition dynamics. The DHTM produces more accurate n-step Successor Features than LSTM and RWKV, which speeds up n-step TD learning of the SF in Reinforced Learning tasks with the changing environment.

One of the limitations of the DHTM is that its temporal context is random, as it is formed on the fly. That is, the mechanism of context formation doesn't allow generalizations. That is why we are forced to use feature space inferred from observations for value function decomposition, to soften this problem. Nevertheless, we believe that forming Successor Features combined with two level hierarchy of DHTM layers may provide the next step to circumvent this limitation, which directs of our further research. Another limitation is the maximum number of variables per factor. The amount of segments in use grows exponentially with the number of variables per factor, especially in noisy environments. Solving this issue would require to modify segment excitation or growth algorithms.

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

## A  ENCODING AND DECODING OBSERVATIONS

Because our model is designed to work with sparse distributed representations and the testing environments do not provide observations as SDRs by default, an encoding procedure is required. For this task, we use a modified version of the Spatial Pooler (SP) (Cui et al., 2017; Mnatzaganian et al., 2017), a distributed noise-tolerant online clustering neural network algorithm that converts input binary patterns into SDRs with fixed sparsity while retaining pairwise similarity (Kuderov et al., 2023). The SP algorithm learns a spatial specialization of neurons' receptive fields using the local Hebbian rule and k-WTA ($k$ winners take all) inhibition (Oster et al., 2009). Here we outline the main differences from the "vanilla" version of the SP algorithm described in Cui et al. (2017).

During an agent's decision-making process pipeline, the SP encoder accepts a current observation $o$ and transforms it to a latent state SDR $z$. In terms of processing, our SP encoder functions as a standard artificial neural network with a k-WTA binary activation function.:

$$\text{overlaps}_i = \beta_i W_i o \tag{10}$$

$$z_i = \mathbb{I}\left[i \in \text{kWTA}(\text{overlaps})\right], \tag{11}$$

where $o$—a binary observation vector, $W_i$—a row-vector representing $i$-th neuron's connection weights (where non-existing connections have zero weights), overlaps$_i$—a value representing the strength of the input pattern recognition with the neuron $i$ [1], $\beta_i$—an $i$-th neuron boosting value, $z_i$—an $i$-th bit of an output SDR, $\mathbb{I}[\ldots]$—an indicator function, kWTA—a $k$-winners-take-all activation function returning $k$ indices of the neurons with the highest overlap.

One difference between the "vanilla" SP algorithm and ours is that we do not distinguish between potential and active neural connections. Because all [existing] connections are active, they all participate in calculating overlaps. In the overlaps calculation, non-binary, that is, real-valued weights are used, similar to artificial neural networks, as shown in equation 10. Furthermore, each neuron has a fixed capacity to produce neurotransmitters, which it distributes between its synaptic connections. This means that we keep all neuron weights normalized and summed to one. While it achieves the same Hebbian learning with homeostatic plasticity as the original SP, the exact formula is slightly different:

$$\tilde{W}_i = W_i + \alpha z_i \frac{\text{RF}_i \odot o}{\sum_j \text{RF}_i \odot o}$$

$$W_i \leftarrow \frac{\tilde{W}_i}{\sum_j \tilde{W}_i}, \tag{12}$$

where $\tilde{W}_i$—a row of new $i$-th neuron weights before normalization, $\alpha$—learning rate, $z_i$—a binary value representing the current activity state of the $i$-th neuron, $RF_i$—an $i$-th row of the binary connectivity matrix representing an $i$-th neuron receptive field, $\odot$—elementwise product, $o$—a binary observation vector.

The original SP algorithm has several drawbacks, including encoding instability caused by an innate homeostatic plasticity mechanism known as boosting, which helps neurons specialize and increases overall adaptability but makes memorization tasks more difficult, and slow processing on large inputs such as images, where an encoding overhead becomes noticeable when compared to overall model timings around 1k input size.

The introduction of the newborn stage, which follows the ideas proposed in Dobric et al. (2022), solves an encoding instability problem. The newborn stage of a spatial pooler occurs during the early stages of its learning process, when its neurons are expected to specialize. The boosting, which is intended to aid in the specialization process, is activated only during the newborn stage and its scale gradually decreases from the configured value to zero. Boosting remains turned off during an encoder's "adulthood", reducing the possibility of spontaneous re-specialization.

To reduce processing overhead, we use a much more sparsified connection matrix than in the original SP version. We randomly initialize connections with 40-60% sparsity, which is typical for the "vanilla" SP. Then, during the newborn stage, we gradually prune the vast majority of the weakest connections, resulting in neurons that are highly specialized due to their small receptive fields. We typically configure the final receptive field size in relation to the average input pattern size (usually 25-200% of it, resulting in 0.1-10% connections sparsity). For example, if binary input patterns have on average 100 active bits out of 1000, we can set the target size of receptive fields to 25, which is 25% of the active input size and corresponds to 2.5% connection matrix sparsity. As a result, the spatial pooler's instability (and thus adaptiveness!) becomes even more limited in the adult stage.

Because of its soft discretization (from the distributed representation) and clusterization properties, we expect SP to assist the model with input sequence memorization and an environment transition dynamics generalization tasks in addition to the encoding itself. However, because the SP encoder learns online, particularly during the newborn stage, its output representation can be highly unstable during the early stages, potentially resulting in a performance drop.

To visualize and debug an encoded observation, we also learn a decoder, which is a linear neural layer learned locally with gradient descend on the MSE error between the predicted reconstruction and the actual observation.

---

[1]While the name "overlap" does not exactly reflect its meaning in our SP modification, because it is not a binary overlap between a receptive field and an input pattern, we kept it on purpose to refer to the similar term commonly used for the original SP.

## B  VALUE FUNCTION DECOMPOSITION

In our agent model, we approximate the reward function R(s) as a sum: $R_t = \frac{1}{n}\sum_{k=1}^{n} r(\varphi_t^k)\varphi_t^k$, where $r(\varphi_t^k)$ is a reward associated with state $\varphi_t^k$, $n$–number of feature variables. Then, similarly to the Successor Representation idea (see Section 2.3), the value function can be represented as:

$$V(h_t) = \mathrm{E}[\sum_{l=0}^{\infty} \gamma^l R_{t+l+1} \mid h_t] = \sum_{l=0}^{\infty} \gamma^l \mathrm{E}[\frac{1}{n}\sum_{k=1}^{n} r(\varphi_t^k) \mid h_t]$$

$$= \frac{1}{n}\sum_{k=1}^{n}\sum_{j}\sum_{l=0}^{\infty} \gamma^l p(\varphi_{t+l+1}^k = j \mid h_t^k) r_j^k$$

$$= \frac{1}{n}\sum_{k=1}^{n}\sum_{j} M_j^k(h_t^k) r_j^k, \tag{13}$$

where $M_j^k(h_t^k) = \sum_{l=0}^{\infty} \gamma^l p(\varphi_{t+l+1}^k = j \mid h_t^k)$, $h_t = (h_t^1, ..., h_t^n)$—hidden state vector of variables $\{H_t^k\}_k$.

Then, the temporal difference for $M_{ij}^k = M_j^k(h_t^k = i)$ is:

$$\delta_{ij}^k = \sum_{l=0}^{T} \gamma^l p(\varphi_{t+l+1}^k = j \mid h_t^k = i)) + \gamma^{T+1}\sum_{m} M_{mj}^k p(h_{t+T+1}^k = m \mid h_t^k = i) - M_{ij}^k, \tag{14}$$

However, in POMDP we can't observe $h_t^k$, we only have a distribution $p(h_t^k \mid o_{0:t})$. Therefore, we need to average out the hidden state variable $\delta_j^k = \sum_i \delta_{ij}^k \cdot p(h_t^k = i \mid o_{0:t})$. Assuming that we minimise $L = (\delta_j^k)^2$, we get the following update rule:

$$\delta_j^k = \mathrm{gen}_{t+T}(\varphi^k = j \mid o_{0:t}) + \gamma^{T+1}\mathrm{pred}_{t+T+1}(\varphi^k = j \mid o_{0:t}) - \sum_i M_{ij}^k p(h_t^k = i \mid o_{0:t}) \tag{15}$$

$$M_{ij}^k \leftarrow M_{ij}^k + \alpha\delta_j^k \cdot p(h_t^k = i \mid o_{0:t}), \tag{16}$$

where $\mathrm{gen}_{t+T}$—Successor Features component, generated by temporal memory up to timestep $T$, and $\mathrm{pred}_{t+T+1}$—SF component predicted using $M_{ij}^k$ weights:

$$\mathrm{gen}_{t+T}(\varphi^k = j \mid o_{0:t}) = \sum_{l=0}^{T} \gamma^l \sum_i p(\varphi_{t+l+1}^k = j \mid h_t^k = i)p(h_t^k = i \mid o_{0:t}) \tag{17}$$

$$\mathrm{pred}_{t+T+1}(\varphi^k = j \mid o_{0:t}) = \sum_i M_{ij}^k p(h_{t+T+1}^k = i \mid o_{0:t}) \tag{18}$$

## C  BIOLOGICAL INTERPRETATION

Neural implementation of the DHTM is inspired by neocortical neural networks (see Fig. 6). Hidden variables $H^k$ may be considered as populations of excitatory pyramidal neurons in cortical layer L2/3 of somatosensory areas, with lateral inhibition modelled as $\mathrm{softmax}$ function. Staiger & Petersen (2021) showed that neurons in this layer are responsible for temporal context formation.

The neuronal activity at timestep $t$ can be thought to carry messages $M_{t-1}^k$. Messages are propagated through synapses of dendritic segments, which correspond to factors $F_c^k$. Dendritic segments of biological neurons are known to be coincidence detectors of its synaptic input (Stuart & Spruston, 2015). We use the notion of dendritic segment to sparsely represent context factors $F_c^k$, as each factor value corresponds to a particular combination of states (or active cells).

Feature variables $\Phi_t^k$ may be considered to represent cells of a granular layer (L4), as they are known to be the main hub for sensory excitation for L2/3. L2/3 cells that have common sensory input from the layer L4 are modelled as columns for particular feature states $\mathrm{col}(\varphi_t^k)$ (Mountcastle, 1997).

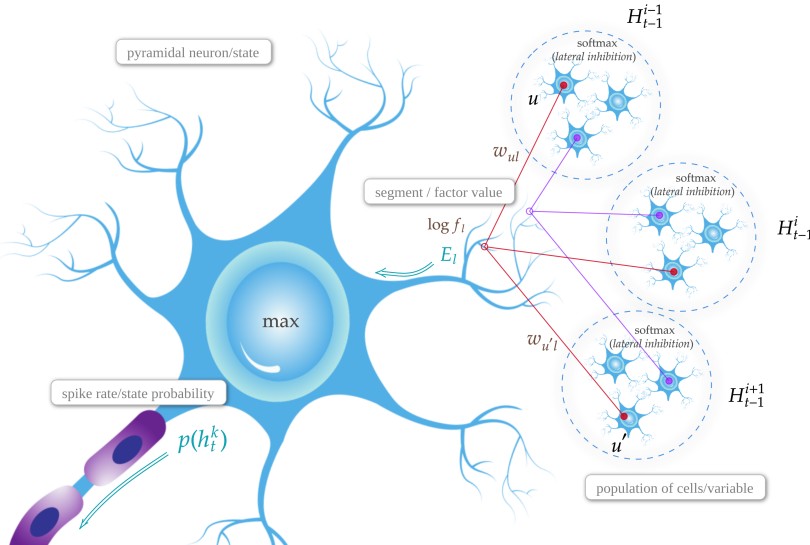

Figure 6: Biological view of the neural implementation of the DHTM. Variables $H_{t-1}$ correspond to populations of neurons that have common sensory input and lateral inhibitory competition. Dendritic segments correspond to factor values $f_l$. spike frequency of a neuron reflects state probability $p(h_t^k)$ of a variable.

## D    PSEUDO-SURPRISE

To calculate the pseudo-surprise of SF, we do the following:

1. Normalize SF by summing it over corresponding variables $\Psi$ and dividing SF by these sums. The result is SF-induced probability distribution $p(\varphi^k)$ of $\Psi^k$ variables.

2. Measure average surprise over future observed states $\varphi^k$, according to this distribution: $-\log p(\varphi^k = j)$, where $j$ is the observed state.

Normalized SF represents future observation (feature) profile for the current state. Pseudo-surprise shows whether SF is consistent with the observed feature states or not. For example, if SF doesn't predict feature $j$ ($p(\varphi^k = j) = 0$), but we observe it, this'll result in infinite surprise, which means that the SF is of a bad quality.

## E    BASELINE RL AGENTS IMPLEMENTATION

As mentioned in Section 3.3, we incorporate DHTM as a part of an RL agent, which has a memory model, an SF mapping from hidden space, and a feature reward function. Our memory model is expected to speed up SF learning for an agent. We put this hypothesis to the test by experimenting with other memory models while keeping the agent architecture the same. Thus, all tested memory models work in the same regime—they learn sequences of encoded binary observations (i.e. SDRs that we get from Spatial Pooler encoder described in Appendix A) concatenated with one-hot encoded actions.

LSTM baseline was implemented with a single LSTMCell from PyTorch library (Paszke et al., 2019). It is supported by an additional symexp-layer to encode input before passing it to the LSTM cell and a symexp-layer to decode the LSTM cell's output from the LSTM's hidden state back to the input representation, where symexp activation function, $\text{symexp}(x) = \text{sign}(x)e^{|x|-1}$, is a reverse of symlog function: $\text{symlog} = \text{sign}(x)\log(|x| + 1)$.

The similar way we implemented RWKV baseline: a single RWKV layer supported by single-layer linear encoder and decoder. Current public RWKV implementation is a fast evolving framework (Bo, 2021), and for the increased performance it is tightly bound to the offline batch training common

for the transformer architectures. In our case we needed a so-called sequential mode for online learning similar to LSTM. Thus, we adapted another public implementation mentioned in the official documentation (RWKV in 150 lines of code).

Both RNNs were trained online with backpropagation through time (BPTT) on the observed sequences with the backpropagation update step scheduled every $k$ timesteps. We experimented with different schedules and found that $k = 20$ provides a balance between the training stability and speed. The learning rate was set to $\alpha = 6 \cdot 10^{-4}$ for LSTM and $\alpha = 5 \cdot 10^{-4}$ for RWKV.

We also incorporated some notion of random variables and their states by splitting the hidden state of the tested RNNs into groups. In all experiments the hidden state represents 80 categorical variables with 4 states. That is, both RNNs are forced to learn 80 categorical distributions with multi-cross-entropy loss to explain the observed sequences, which is a somewhat close to the multi-categorical hidden state representation used in DreamerV2/V3 (Hafner et al., 2023). The idea of using symexp activation function, mentioned above, is inspired by Dreamer too, and is used to remedy the problem of learning extreme probability values. Without symexp the neural network has to represent zero probability with high negative logit values and one-probability with high positive logit values, which is hard to reach with low learning rate and may lead to instabilities. Thus, symexp function makes it faster to reach target values in log space.

CSCG baseline was implemented using code from the repository accompanying the paper (`https://github.com/vicariousinc/naturecomm_cscg`). In our experiments, in order to handle multiple feature variables, we trained several CSCGs independently using the same data. CSCG was trained on batches with size of 500 observation steps. We iteratively calculated exponential moving average of transition matrices obtained for different batches with smoothing coefficient $\alpha = 0.8$. This smoothed transition matrix was used as initialization for the next batch training and for inference.

All baselines employed a multilayer perceptron, implemented with PyTorch, in order to map from the hidden state distribution to Successor Features because the simple linear model described in Appendix B didn't work for them. The MLP had one hidden layer with size 256 units and batch size of 32 for CSCG and 256 for LSTM and RWKV with squared temporal difference as a loss function defined by equation 16.

## F    EXPERIMENTAL SETUPS

Pinball is a partially observable environment developed in the Godot Game Engine (Beeching et al., 2021). A ball that can move in the surface's 2D space and a surface with borders make up the environment (see Figure 7-A). Force fields depicted as circles introduce stochasticity to the environment as they deflect the ball in random directions. An agent can apply arbitrary momentum to a ball. For each time step, the environment returns an image of the top view of the table as an observation and a reward. The agent gets the reward by entering force fields. Each force field can be configured to pass a specific reward value and to terminate an episode.

AnimalAI is a testbed inspired by experiments with animals (Crosby et al., 2020). The environment consists of 3D area surrounded by a wall and many different objects that can be placed using a configuration file including: walls, food, ramps, trees, movable obstacles and so on (see Figure 7-C).

## G    2D MAZE VALUE FUNCTIONS

## H    GLOSSARY

**Categorical Random Variable**—a discrete random variable that can take on of finite $K$ possible states.
**Cortical Column or Minicolumn**—a population of neurons in the neocortex that spans across layers and shares sensory input.
**Dendritic segment**—a group of synapses (neuron's connections) that acts as an independent computational unit affecting the resulting neuron's activity.

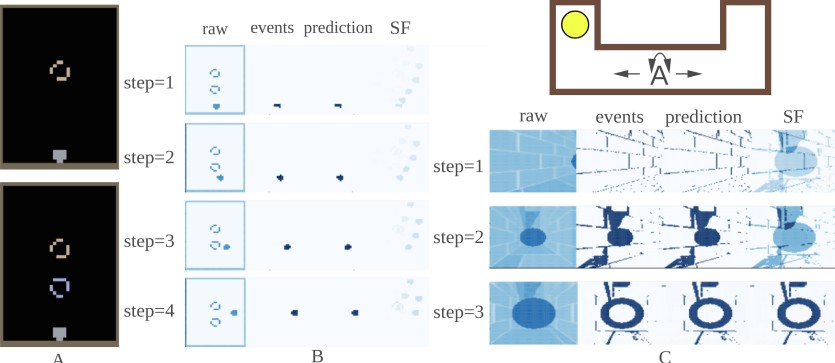

Figure 7: A. Pinball experiments used two different setups. The upper image shows a setup in which the target is not blocked. The lower image depicts the setup, with the target obscured by a random field that deflects the ball perpendicular to its movement direction. B. Visualization of several steps in the Pinball environment. Each step is depicted by raw observation image, binary image of events, predicted events and Successor Features. C. Animal experimental setup: two corridors, one of which containing food (yellow circle), the agent is in between of the corridors (letter A). Food position changes after 300 episodes. Images on the right: observations (raw), processed observations (events), predictions and Successor Features decoded back to observation space for three last steps of an episode.

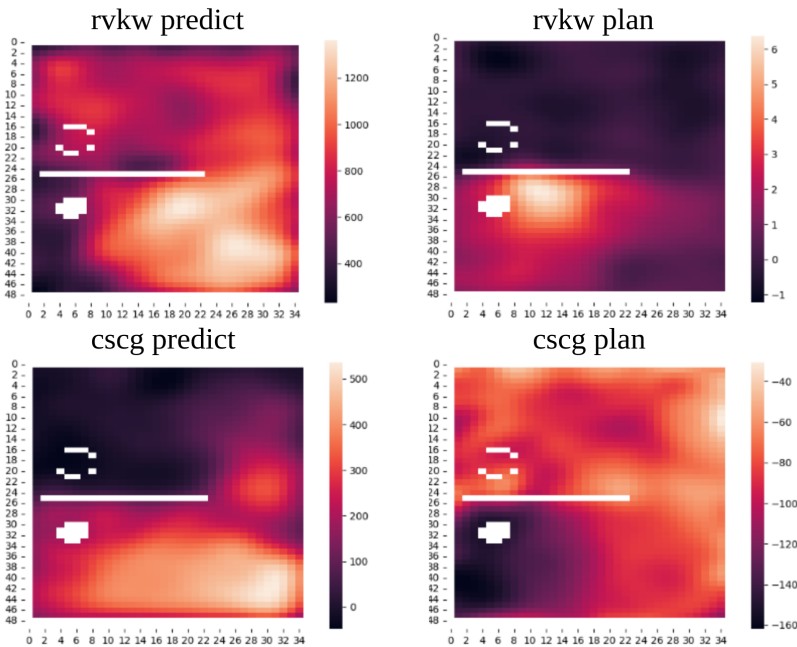

Figure 8: Heatmaps representing value function in 2D maze Pinball environment setup.

**Factor Graph**—bipartite graph representing the factorization of a probability distribution, with one part representing factor nodes and another—random variables.

**Multi-compartment neuron model**—a model of neuron that divides neuron's connections into groups (segments) of different types (compartments), where each group may be considered as partly independent computational unit and groups of each compartment may affect the neuron's activity differently.

**Sparse Distributed Representations (SDR)**—sparse binary vector in a high-dimensional space, usually formed by k-WTA algorithms.

**Spatial Pooler (SP)**—a distributed noise-tolerant online clustering neural network algorithm that converts input binary patterns into SDRs with fixed sparsity while retaining pairwise similarity.
**Successor Representations (SR)**—a discounted sum of future [one-hot encoded] observations.
**Successor Features (SF)**—a generalization of SR, a discounted sum of future latent states.
**Temporal Memory (TM)**—in this work by this term we mean "memory for sequences".
**Hidden Markov Model (HMM)**—statistical model of a stochastic process where state probability depends only on previous state of the process.

