# OpenReview forum: "Learning Successor Representations with Distributed Hebbian Temporal Memory"
_ICLR.cc/2024/Conference — Submitted to ICLR 2024_

### Official Review · Reviewer_kwS4 · 2023-10-26

**Soundness:** 1 poor
**Presentation:** 3 good
**Contribution:** 2 fair
**Rating:** 3
**Confidence:** 4

**Summary:**

The current work implements belief propagation in Hidden Markov Models using neural circuits.

**Strengths:**

The implementation of belief propagation in a neural circuit is innovative.

**Weaknesses:**

The article has three main shortcomings:

There is a lack of theoretical analysis for the proposed algorithm. Many parts of the article involve approximations or assumptions, but the author fails to discuss these aspects. For example, the author does not explain why the 'independent' variable can be represented in the form of Eq. (11), and the parameters such as w_ul, learned using Hebbian rules, lack clear explanations. The learning targets and the specific characteristics of the learning dynamics are also left unaddressed.

Secondly, the author fails to provide a clear description of the entire workflow of the algorithm. It is uncertain whether the sum–product message passing can converge within the given graphical structure. If it does converge, it needs time to do so. It is unclear how many iterations are required for message m_u to converge. If multiple iterations are needed for convergence, the transmission of information from time t+1 to t may be questionable. Actions also affect the transition matrix of HMM, but in Section 3.1, I did not see the author modeling actions.

The article does not clarify the specific biological correspondences of 'f' and 'm' within the context of sum–product message passing. 'f' appears to correspond to neural connections, but neural connections are typically one-to-one. If 'f' includes more than two nodes, its precise correspondence within a neural circuit is unspecified. Similarly, the biological interpretation of 'm' is not clearly defined, and the mechanism for nodes to transmit 'm' among each other remains unclear.

**Questions:**

See weakness

---

> ### Author Response · Authors · 2023-11-17
> **On Theoretical Analysis and Biological Interpretation of the Work**
>
> Thank you for your valuable comments and questions. They shed light on some of the presentation weaknesses of the work, we have added corresponding clarifications that improved the overall paper’s quality.
>
> ### W1 Segment Likelihood Approximation
> > the author does not explain why the 'independent' variable can be represented in the form of Eq. (11)
>
> **Our response:** As we mentioned in the paper, the Eq. (11) is a linear approximation of log-likelihood of segment activation between two extreme cases. We don’t know if presynaptic variables are independent or not from the outset, that’s why we learn synaptic weights $w_{ul}$, which correspond to $p(s_l=1 | c^u_{t-1}=1)$ – probability that segment $l$ is active, given cell $u$ is active. We can estimate $w_{ul}$ empirically, just by counting coincidences of segment and presynaptic cell activities. Segment $l$ learns to respond to a cluster of active cells, that’s why if $w_{ul} \to 1$, that means cell $u$ is clustered, otherwise it is independent of the cluster that activates segment $l$.
>
> ### W2 Sum-product Convergence
> > It is uncertain whether the sum–product message passing can converge within the given graphical structure.
>
> **Our response:** There are no loops in our graphical structure, so it needs only one iteration of message passing (see Figure 1A).
>
> ### W3 Modelling Actions
> > Actions also affect the transition matrix of HMM, but in Section 3.1, I did not see the author modeling actions.
>
> **Our response:** Thank you for commenting on that, we have added corresponding clarifications to the text. We include actions in the model by forcing some of the variables $H^k_{t-1}$ to represent actions. That is, we assume that information about action is included in the hidden state of the model. If we have 4 actions, we set 4 states for one of the hidden variables and set its state from observation of the action.
>
> ### W4 Biological Interpretation
> > The article does not clarify the specific biological correspondences of 'f' and 'm' within the context of sum–product message passing.
>
> **Our response:** As we mentioned in the Section 3.1, Factor value $f$ corresponds to some learnable parameter of a dendritic segment. We speculate, that $f$ is segment’s efficacy, i.e. how strong the contribution of the segment to cell’s spiking is. It’s known that a segment may have several presynaptic cells and can be considered as a coincidence detector [1], from where factors with several variables come. Value $m$ is just normalized spike frequency of the corresponding cell. We agree, that it’s not clear from the text, so we added a separate section for clarifying biological interpretation of our model.
>
> [1]  Stuart, Greg J., and Nelson Spruston. 2015. ‘Dendritic Integration: 60 Years of Progress’. Nature Neuroscience 18 (12): 1713–21. https://doi.org/10.1038/nn.4157.

---

> > ### Comment · Reviewer_kwS4 · 2023-11-22
> >
> > Thanks for your detailed answer. As other reviewers have pointed out, the article involves a lot of theory and is a great challenge in writing. I believe there is still much room for improvement.

---

### Official Review · Reviewer_aAVr · 2023-11-01

**Soundness:** 2 fair
**Presentation:** 2 fair
**Contribution:** 2 fair
**Rating:** 3
**Confidence:** 3

**Summary:**

This paper introduces Distributed Hebbian Temporal Memory (DHTM), a state-update function used to derive an approximate Markov state in partially observable environments. The DHTM is modeled as a factor graph of an HMM with additional modifications to consider the dependence of hidden states. The paper’s primary contribution is to introduce a graphical model that approximates the successor representation in finite horizon POMDPs.

**Strengths:**

- The problem that the authors address is an important one, i.e., developing models that can infer the underlying Markov state from partially observed sequential data.
- DHTM relies on local Hebian-like updates, avoiding some of the issues of gradient-based sequence models employing backpropagation like RNNs.

**Weaknesses:**

I think the current state of the paper isn’t in a form I would feel comfortable recommending for acceptance. The paper does present a novel architecture, but much of the paper relies on empirically demonstrating DHTM’s effectiveness, which I believe the authors haven't done convincingly. Furthermore, I feel the authors should discuss the limitations of their method and better discuss related work. Below, I’ll address specific concerns:

- The method as presented wouldn’t scale to infinite horizon or large POMDPs. At the very least, I’d like to see a discussion on the challenges presented when scaling DHTM.
- The empirical methodology and agent architecture are poorly explained, which makes it hard to evaluate the method. More direct questions can be found below.
- The empirical results are of limited value; specifically, the authors use a single custom environment with non-standard metrics (see below for more questions about pseudo-surprise).
- The paper could be better positioned in the literature regarding successor representations in POMDPs. For example, I expected to see a discussion with [1], which seeks to model an SR in POMDPs.

[1] Eszter Vertes and Maneesh Sahani. A neurally plausible model learns successor representations in partially observable environments. NeurIPS 2019.

**Questions:**

- Section 2.1, shouldn't optimal policy be defined as $\text{argmax}_a$ not $\text{softmax}_a$?
- The SR in RL is defined with respect to a policy; in DHTM, the target policy is ignored. Is this on purpose? If so, could authors comment on this and the difficulties of adapting their estimate with a changing policy?
- Section 2.3, I’m quite confused about this section; although the SR could be used to model observations, you’ve discussed the SR up to this point as being a good representation of the hidden state. I’m not sure that the expected discounted observation occupancy, given a state, is the same object. It feels like successor features should be discussed here, as there’s an implicit assumption that the reward function lies in the span of the observation features. You might not be able to linearly predict the value function $V(h)$ given $M^\pi(\cdot | h) \in \mathbb{R}^{|\mathcal{O}|}$.
- Section 4, can you describe how the pseudo-surprise is computed? I’m not sure I fully understand what’s going on here.

### Agent Architecture

- Can you provide more details about the baseline methods and how they are employed in the architecture? e.g., what’s the input to the RNNs?
- What is an SR representation here? A tabular representation of the SR with respect to the HMM states?

---

> ### Author Response · Authors · 2023-11-20
>
> Thank you for your valuable feedback. We took into consideration all our comments and will make corresponding adjustments and extensions to the paper ASAP, and we will notify you about the changes. We hope that this will raise the significance of our inquiry and the quality of the presentation of our algorithm.
>
> ### Q1 Policy Distribution
> > Section 2.1, shouldn't optimal policy be defined as $\text{argmax}_a$ not $\text{softmax}_a$?
>
> **Our response:** Thank you noticing that! Our algorithm employs softmax strategy for enabling exploration, however, indeed, the optimal strategy is deterministic.
>
> ### Q2 Including Actions Into the Model
> > The SR in RL is defined with respect to a policy; in DHTM, the target policy is ignored. Is this on purpose? If so, could authors comment on this and the difficulties of adapting their estimate with a changing policy?
>
> **Our response:** Thank you for commenting on that, we will add corresponding clarifications to the manuscript soon. We include actions in the model by forcing some of the variables $H^k_{t-1}$ to represent actions. That is, we assume that information about action is included in the hidden state of the model. If we have 4 actions, we set 4 states for one of the hidden variables and set its state from observation of the action. We form on-policy SRs, i.e. relying on policy iteration theorem. DHTM can be used to form SRs for any policy, given it learned dynamics of the environment.
>
> ### Q3 Successor Features instead of Successor Representations
> > Section 2.3, I’m quite confused about this section; although the SR could be used to model observations, you’ve discussed the SR up to this point as being a good representation of the hidden state. I’m not sure that the expected discounted observation occupancy, given a state, is the same object. It feels like successor features should be discussed here, as there’s an implicit assumption that the reward function lies in the span of the observation features. You might not be able to linearly predict the value function $V(h)$ given $M^\pi(\cdot | h) \in \mathbb{R}^{|\mathcal{O}|}$.
>
> **Our response:** Thank you for noticing that! We have called it Successor Representation as a widespread term among neuroscientists (see [1] for example), and we aimed to bridge knowledge about place cells to our work. Nevertheless, we agree that it’s not common to define Successor Representation in observation space, and this may cause confusion for the reader familiar with the topic from the RL field. We will reformulate our work in terms of the Successor Features framework. In DHTM, variables $O$ are encoded observations, so they are, in effect, already in feature space, not raw observations from the environment. Variables $O$ are observations with respect to the memory itself, but not to the agent. To avoid confusion, further, we call these variables $\Phi$ to distinct them from raw observations.
>
> [1] Samuel J. Gershman. 2018. ‘The Successor Representation: Its Computational Logic and Neural Substrates’. The Journal of Neuroscience 38 (33): 7193. https://doi.org/10.1523/JNEUROSCI.0151-18.2018.
>
> ### Q4 Custom Metric: Pseudo-Surprise of SF
> > Section 4, can you describe how the pseudo-surprise is computed? I’m not sure I fully understand what’s going on here.
>
> **Our response:** As we mentioned in the text, to measure pseudo-surprise of SF, we do the following:
> 1. Normalize SF by summing it over corresponding variables $\Phi$ and dividing SF by these sums. The result is SF-induced probability distribution $p(\varphi^k)$ of $\Phi^k$ variables.
> 2. Measure average surprise over future observed states $\varphi^k$, according to this distribution: $-\log p(\varphi^k=j)$, where $j$ is the observed state.
>
> Normalized SF represents future observation (feature) profile for the current state. Pseudo-surprise shows whether SF is consistent with the observed feature states or not. For example, if SF doesn’t predict feature $j$ ($p(\varphi^k=j) = 0$), but we observe it, this’ll result in infinite surprise, which means that the SF is of a bad quality. Thank you for your comment, we will devote more space to this explanation in the text. For comparison, we will also put this metric calculated for uniform SFs.

---

> ### Author Response · Authors · 2023-11-20
>
> ### W1 Scalability of the method
> > The method as presented wouldn’t scale to infinite horizon or large POMDPs. At the very least, I’d like to see a discussion on the challenges presented when scaling DHTM.
>
> **Our response:** The main idea of our agent architecture is to combine planning and TD learning, which results in n-step TD learning. It doesn’t need infinite horizon predictions as it learns linear function mapping from hidden space to Successor Features space. We only need to predict the first several steps — to speedup TD learning and reduce its bias — and approximate the tail with the learned mapping (see Eq. (4)).
>
> Scaling to POMDPs with large observation space is also possible due to usage of multiple feature variables, which constitute sparse distributed representation of observations. Sparsity [of observations and connections] is the important key to deal with high dimensionality in our approach. To show that, we also added experiments in AnimalAI environment, which gives more complicated visual input to the agent.
> We should have stated it clearer, as scalability is what we aimed for designing DHTM. Nevertheless, our method still may have scalability issues, so we will add corresponding discussion to the Conclusion section. Thank you for pointing this out.
>
> ### W2 Architecture Description
> > The empirical methodology and agent architecture are poorly explained, which makes it hard to evaluate the method. More direct questions can be found below.
>
> **Our response:** We agree that the agent’s architecture could be better explained. We took your comment into consideration and will extend the description of the agent and the experimental setup.
>
> ### W3, W4 Lack of Empirical Results
> > The empirical results are of limited value; specifically, the authors use a single custom environment with non-standard metrics (see below for more questions about pseudo-surprise).
>
> > The paper could be better positioned in the literature regarding successor representations in POMDPs. For example, I expected to see a discussion with [1], which seeks to model an SR in POMDPs.
>
> **Our response:** We agree that our custom environment experiments may be not very convincing, so we will add results of experiments in AnimalAI, which is a known benchmark for Reinforcement Learning architectures [2]. Besides, we are going to add standard n-step surprise metrics and qualitative examples of the resulting value functions. We have also tested our algorithm in a 2D maze experiment, similar to what is used in the paper you mentioned, and will add the results to the paper soon.
>
> [1] Eszter Vertes and Maneesh Sahani. A neurally plausible model learns successor representations in partially observable environments. NeurIPS 2019.
> [2] Crosby, Matthew, Benjamin Beyret, and Marta Halina. 2019. ‘The Animal-AI Olympics’. Nature Machine Intelligence 1 (5): 257–257. https://doi.org/10.1038/s42256-019-0050-3.

---

> ### Author Response · Authors · 2023-11-23
> **Updates**
>
> We have uploaded an updated version of the text. We have added baselines description to the Appendix and formulation of value function decomposition we use.

---

### Official Review · Reviewer_wRcd · 2023-11-04

**Soundness:** 2 fair
**Presentation:** 1 poor
**Contribution:** 2 fair
**Rating:** 3
**Confidence:** 3

**Summary:**

This paper introduces an novel approach for learning sequences in partially observable environments called Distributed Hebbian Temporal Memory (DHTM). DHTM is rooted in factorial HMMs and incorporates local Hebbian-like learning rules and transition matrix sparsification, both inspired by biologically plausible multi-compartment neural models. The authors illustrate DHTM's capability to learn successor representations and demonstrate its functionality in a simple pinball environment.

**Strengths:**

This paper tackles the significant challenge of online hidden representation learning for decision-making in non-stationary, partially observable environments. The proposed method, DHTM, amalgamates various intriguing concepts, such as factorial HMMs, successor representations, and multi-compartment neuron models. The biological plausibility of the method is a notable feature. I believe there is ample potential for further exploration and the establishment of very interesting results in this research direction.

**Weaknesses:**

This paper brings together a wide array of interesting concepts, including factorial HMMs, successor representations, multi-compartment neuron models, and sparse distributed representations, making it a dense and challenging read. It would definitely benefit from a longer format, such as a journal article. However, there remains three-quarters of a page that could be employed to break down and refine Section 3.1 on Distributed Hebbian Temporal Memory. Dividing this section into smaller subsections would aid in better organizing the material. Additionally, incorporating a notation glossary, either in the main paper or in an appendix, would be a valuable addition.

Although the model itself is quite compelling, the experiments fall short of doing it justice. Only a single experimental result is provided, where the performance is marginally inferior to one of the baseline models, the fchmm. There is potential for more comprehensive experiments that can showcase all the intriguing aspects of this model.

Minor issues include typos and grammatical errors scattered throughout the text.

**Questions:**

- In section 2.4, when you mention the "spatiotemporal property of cortical network activity," what specific aspect or characteristic are you referring to?
- In the factor graph model, what do the superscripts $i$ and $k$ correspond to?
- What parameter values were used for DHTM for the results in Figures 3 and 4? What about for the other methods?

---

> ### Author Response · Authors · 2023-11-20
>
> Thank you for your valuable commentary and advice! We agree with most of the observations and will do our best responding to them ASAP, and we will notify you about the paper updates. We hope that corresponding adjustments and extensions of our work will improve its quality and significance.
>
> ### Q1 Spatiotemporal Property of Cortical Network Activity
> >In section 2.4, when you mention the "spatiotemporal property of cortical network activity," what specific aspect or characteristic are you referring to?
>
> **Our response:** We are referring to such properties of cortical activity that only a small percentage of neurons are active at each moment of time and the semantics is spread (or distributed) across them.
>
> ### Q2 Graph Model Superscripts
> >In the factor graph model, what do the superscripts $i$ and $k$ correspond to?
>
> **Our response:** Superscripts $i$ and $k$ are used to enumerate different variables as in our model we have multiple hidden and feature variables at the same time, similarly to the Factorial HMM[1].
>
> [1] Ghahramani, Z., and M.I. Jordan. 1997. ‘Factorial Hidden Markov Models’. Machine Learning 29 (2–3): 245–73. https://doi.org/10.1023/a:1007425814087.
>
> ### Q3 Parameter Values for DHTM and Other Methods
> >What parameter values were used for DHTM for the results in Figures 3 and 4? What about for the other methods?
>
> **Our response:** Thank you for pointing this out! We will add the most important parameters’ values of our model and baselines as an Appendix section. All parameters were chosen to optimize the performance of the methods, using grid search among the most plausible parameter values.

---

> ### Author Response · Authors · 2023-11-20
>
> ### W1 Improving Readability of the Text
> >However, there remains three-quarters of a page that could be employed to break down and refine Section 3.1 on Distributed Hebbian Temporal Memory. Dividing this section into smaller subsections would aid in better organizing the material. Additionally, incorporating a notation glossary, either in the main paper or in an appendix, would be a valuable addition.
>
> **Our response:** Thank you for your advice! We are going to divide Section 3.1 into several subsections and add a glossary at the end of the paper as soon as possible.
>
> ### W2 Lack of Empirical Evidence
> >Although the model itself is quite compelling, the experiments fall short of doing it justice. Only a single experimental result is provided, where the performance is marginally inferior to one of the baseline models, the fchmm. There is potential for more comprehensive experiments that can showcase all the intriguing aspects of this model.
>
> **Our response:** We agree that one experimental result may be not enough to justify the model. Therefore, we have added an easily interpretable 2D maze experiment to show qualitatively that DHTM value function estimate is adequate. Also, we have added examples of the learned Successor Features for our initial setup, which are easily interpretable too. Besides, we have added results of experiments in AnimalAI environment to show that our method is scalable and environment agnostic. We will upload an updated version of the paper ASAP.
>
> Although our method is inferior to FCHMM according to some metrics, we should emphasize that, in contrast to FCHMM and other baselines, our method learns in fully online mode and doesn’t employ error backpropagation algorithm. FCHMM needs the whole sequence being available while updating learnable parameters. So, FCHMM provides a lower [surprise] bound that we seek to achieve, using only online Hebbian learning. However, hidden state representations, resulting from FCHMM are, in contrast to our method, dense, and that hinders linear mapping of hidden space to Successor Features. We will discuss this and other limitations of the baselines in the Conclusion section.
>
> ### W3 Grammatical Errors
> >Minor issues include typos and grammatical errors scattered throughout the text.
>
> **Our response:** Thank you for noticing that! We did another iteration of grammar checking.

---

> ### Author Response · Authors · 2023-11-23
> **Updates**
>
> We have uploaded an updated text with additional experiments in different environments.

---

### Official Review · Reviewer_qUQp · 2023-11-06

**Soundness:** 2 fair
**Presentation:** 2 fair
**Contribution:** 2 fair
**Rating:** 3
**Confidence:** 3

**Summary:**

In this work, the authors proposed a novel approach to learn successor representations inspired by neuroscience. In particular, the model uses an underlying graphical model and applies message passing algorithms. The messages from the latent representations of previous timesteps contribute to the factors which are then used to compute the expected value function of the latent representation and the resulting policy. The authors tested their hypothesis in a 2D pinball environment against LSTM, a transformer based recurrent neural network model (Receptance Weighted Key Value) and a recurrent neural network that has been extended with a linear recurrent unit to handle long sequence modeling tasks (LRU). Considering a metric based on pseudo-surprise, the authors claimed that their model is able to approximate the successor representations better. They also simulated a changing task by comparing successor representations learned using 5-step vs 1-step temporal difference (TD) learning. Unsurprisingly, perhaps due to better reward error propagation, the model that uses the 5-step TD learning algorithm is able to adapt to the second task faster.

**Strengths:**

Overall, the writing and presentation are straightforward. The authors kept the writing simple which made it easier for me to digest the paper. The motivation of pursuing the research idea is clear.

**Weaknesses:**

The figures and the structure of the paper can be greatly improved. Please see the questions below.

The authors only consider one simple environment. It will be interesting to study their approach in 2D maze, such as minigrid environments which are commonly used to study successor representations. This environment will also allow more analysis and make better and more accurate comparisons with ground truth successor representations.

At the moment, it is hard to understand why their proposed model is performing better than the baseline. The baseline models seem to have more complex architectures but no learning curves were presented to verify that the complexity reduces the learning efficiency of these models.

**Questions:**

1.  Writing: Please spell out the full term at its first mention, indicate its abbreviation in parenthesis and use the abbreviation from then on. Example: Receptance Weighted Key Value (RMKV), Hidden Markov Model (HMM) etc. It makes it much easier to read and follow, especially for those who might not be familiar with all these models.
2. There are some grammar errors:
   a. … which can be formalized as agent?? Reinforcement Learning for a Partially Observable Markov Decision Process (POMDP) (Poupart, 2005).
3. Avoid using M for both successor representations and Messages. I understand that it is common to use M for SRs. Perhaps you can use “m” for messages.
4. The caption for Figure 1 is lacking in detail and doesn't provide enough context to fully grasp its significance. It's understandable that you wouldn't want to repeat the main text, but some additional details in the caption would enhance clarity. The authors might also want to consider adding equations or indicating the link to the equations in the figure.
  a .1. What is i here?
  b. Figure B is unclear. What do the variables E represent? Likewise, are the w_u and w_u'_l? 4. What is E?
  c.What does the red circle represent in Figure C? Is Figure C the overall graphical model?

5. It was stated in the abstract that the proposed model is inspired by “neurophysiological models of the neocortex, distributed representations, sparse transition matrices, and local Hebbian-like learning rules.” The introduction of binary cell activities, presynaptic cells as well as receptive fields could be greatly enhanced by including them in Figure 1. This would create a more cohesive and complete visual representation of the concepts.

6. It is a common practice to use mathbb{I} for indicator function. This will help to remove the confusion with the Identity matrix.

7. In algorithm 1, link it with the equations being optimized during the learning process. At the moment, it is hard to tell how your proposed algorithm contributes to the standard reinforcement learning framework.

8. “The next step after computing p(h_{t}^{k}) distribution parameters is to incorporate information about .."  How does this relate to algorithm 1?

9. How are the weights w_u_l being learned? Is it through equation 18? If it is, eq 18 should be moved to the main paper.

10. Details for the agent’s architecture are missing. Please provide them in the appendix. Ideally, given that you proposed a novel method, a computational graph of your architecture should also be included so that the readers can relate it to figure 1 and algorithm 1.

11. Figure 4 seems to have very high variance. How many seeds are being used to generate the results in this plot? Looks like more seeds are required.

12. Also, in figure 4, there seems to be a stability issue for the 1 step model as the average reward did not stabilize for the first task and starts to diverge before the task switches.

---

> ### Author Response · Authors · 2023-11-20
>
> We are grateful for your thorough commentaries on our paper. We think they have helped us to improve the paper significantly. An updated version of the paper will be uploaded as soon as possible, and we will inform you about the changes.
>
> ### Q1, 2, 3, 6 Minor Text Improvements
>
> **Our response:** Thank you for such a detailed commentary! We took into account your suggestions and observations, which have helped us to improve the overall paper's quality.
>
> ### Q4, 5 Enhancing Clarity of Figure 1.
>
> **Our response:** We are grateful for your suggestions! We agree that we should have included a more detailed description of the figure either into the main text or caption. Corresponding corrections are made, diagrams are enhanced with additional details to facilitate comprehension. We hope it will improve the overall presentation of the work.
>
> ### Q7 Clarification of Algorithm 1.
> > In algorithm 1, link it with the equations being optimized during the learning process. At the moment, it is hard to tell how your proposed algorithm contributes to the standard reinforcement learning framework. 8.“The next step after computing p(h_{t}^{k}) distribution parameters is to incorporate information about .. How does this relate to algorithm 1?
>
> **Our response:** Algorithm 1 scarcely portrays common training procedure for all methods used, not only for DHTM. Algorithm 1 is not valuable as a contribution, rather it is a clarification of memory testing procedure. There is no strict correspondence between steps described in the Section 3.1 and Algorithm 1, rather it’s a description of the agent's architecture workflow.
>
> ### Q8 Learning Synaptic Weights of a Segment
> >How are the weights w_u_l being learned? Is it through equation 18? If it is, eq 18 should be moved to the main paper.
>
> **Our response:** In Section 3.1 we mention that synaptic segment weights $w_{ul}$ are learning through local Hebbian-like plasticity. They are targeted to represent probability $p(s_l=1\ |\ c^u_{t-1}=1)$ that segment $s_l$ is active, given cell $u$ was active at the previous time-step. It can be learned just by counting activation coincidences and mismatches. But in our algorithm it is approximated as exponential moving average of segment’s $s_l$ frequency activation, given $c^u_{t-1}=1$: $\Delta w_{ul} = \alpha * \mathbb{I}[c^u_{t-1}=1] * (\mathbb{I}[s_l=1] - w_{ul})$, where $\alpha \in [0, 1)$ — learning rate. We will add this explanation to the main text to avoid reader confusion.
>
> ### Q9 Agent Architecture Details
> >Details for the agent’s architecture are missing. Please provide them in the appendix. Ideally, given that you proposed a novel method, a computational graph of your architecture should also be included so that the readers can relate it to figure 1 and algorithm 1.
>
> **Our response:** Thank you for your interest! We will add a more detailed description of the algorithm to an Appendix.
>
> ### Q10, 11 Noisy Results
> >Figure 4 seems to have very high variance. How many seeds are being used to generate the results in this plot? Looks like more seeds are required.
>
> **Our response:** As we mentioned in Section 4, we tested three seeds for every setup. We agree that, perhaps, we need to increase the number of trials to show stable results. However, we should note that high variance may be inevitable, due to usage of non-deterministic softmax strategy. We will update experiments and corresponding figures ASAP.

---

> ### Author Response · Authors · 2023-11-20
>
> ### W1 2D Maze Experiments
> >The authors only consider one simple environment. It will be interesting to study their approach in 2D maze, such as minigrid environments which are commonly used to study successor representations. This environment will also allow more analysis and make better and more accurate comparisons with ground truth successor representations.
>
> **Our response:** Thank you for your suggestion! We agree that experiments in 2D maze would be easier to interpret. Therefore, we are going to add them into an updated version of the manuscript.
>
> ### W2 Learning Curves
> >At the moment, it is hard to understand why their proposed model is performing better than the baseline. The baseline models seem to have more complex architectures but no learning curves were presented to verify that the complexity reduces the learning efficiency of these models.
>
> **Our response:** Thank you for pointing this out! We will add learning curves as well.

---

> ### Author Response · Authors · 2023-11-23
> **Updates**
>
> We have uploaded an updated manuscript with additional 2D maze experiments and other improvements you suggested.

---

### Meta-Review · Area_Chair_WngH · 2023-12-04

**Metareview:**

This paper presents a model for learning successor features for an HMM using a form of learning that is similar to the Baum-Welch rule with a Hebbian form. The authors show that the algorithm can be used to learn appropriate features and use those to estimate value functions in 2D Pinball and Animal AI.

The reviewers had several major concerns with this paper. Most notably, several reviewers highlighted a lack of clarity and the empirical experiments being insufficiently convincing. All four reviewers gave a score of 3, which did not change post rebuttal. Given this, a decision of reject was natural.

**Justification For Why Not Higher Score:**

The reviewers were unanimous: this paper is not up to standard for acceptance. Moreover, after looking it over, I personally agree with this assessment. The paper is poorly written and the empirical experiments are underwhelming.

**Justification For Why Not Lower Score:**

N/A

---

### Decision · Program_Chairs · 2024-01-16

Reject